# Alginate: From Food Industry to Biomedical Applications and Management of Metabolic Disorders

**DOI:** 10.3390/polym12102417

**Published:** 2020-10-20

**Authors:** Roxana Gheorghita Puscaselu, Andrei Lobiuc, Mihai Dimian, Mihai Covasa

**Affiliations:** 1Department of Health and Human Development, Stefan cel Mare University of Suceava, 720229 Suceava, Romania; roxana.puscaselu@usm.ro (R.G.P.); andrei.lobiuc@usm.ro (A.L.); 2Department of Computers, Electronics and Automation, Stefan cel Mare University of Suceava, 720229 Suceava, Romania; dimian@usm.ro; 3Integrated Center for Research, Development and Innovation in Advanced Materials, Nanotechnologies, and Distributed Systems for Fabrication and Control, Stefan cel Mare University of Suceava, 720229 Suceava, Romania; 4Department of Basic Medical Sciences, College of Osteopathic Medicine, Western University of Health Sciences, Pomona, CA 91766, USA

**Keywords:** drug delivery, wound dressing, metabolic disorders, microbiome, weight control, probiotics, diabetes

## Abstract

Initially used extensively as an additive and ingredient in the food industry, alginate has become an important compound for a wide range of industries and applications, such as the medical, pharmaceutical and cosmetics sectors. In the food industry, alginate has been used to coat fruits and vegetables, as a microbial and viral protection product, and as a gelling, thickening, stabilizing or emulsifying agent. Its biocompatibility, biodegradability, nontoxicity and the possibility of it being used in *quantum satis* doses prompted scientists to explore new properties for alginate usage. Thus, the use of alginate has been expanded so as to be directed towards the pharmaceutical and biomedical industries, where studies have shown that it can be used successfully as biomaterial for wound, hydrogel, and aerogel dressings, among others. Furthermore, the ability to encapsulate natural substances has led to the possibility of using alginate as a drug coating and drug delivery agent, including the encapsulation of probiotics. This is important considering the fact that, until recently, encapsulation and coating agents used in the pharmaceutical industry were limited to the use of lactose, a potentially allergenic agent or gelatin. Obtained at a relatively low cost from marine brown algae, this hydrocolloid can also be used as a potential tool in the management of diabetes, not only as an insulin delivery agent but also due to its ability to improve insulin resistance, attenuate chronic inflammation and decrease oxidative stress. In addition, alginate has been recognized as a potential weight loss treatment, as alginate supplementation has been used as an adjunct treatment to energy restriction, to enhance satiety and improve weight loss in obese individuals. Thus, alginate holds the promise of an effective product used in the food industry as well as in the management of metabolic disorders such as diabetes and obesity. This review highlights recent research advances on the characteristics of alginate and brings to the forefront the beneficial aspects of using alginate, from the food industry to the biomedical field.

## 1. Introduction

The past decade has experienced a flurry of research in the quest to uncover naturally derived products with unique physicochemical properties and a high degree of compatibility for food and drug delivery applications. Among them, alginate, is the most widely known natural polyanionic polymer, initially used in the food industry to coat fruits and vegetables [1,2,3], for microbial and viral protection [4], and as a gelling, thickening, stabilizing [5] or emulsifying agent [6]. For example, alginate-based films, which are completely biodegradable and edible, have long been used as packaging material for water-soluble powder products such as coffee, coffee-based specialties, powdered milk and instant teas. Likewise, these films have been used in the pharmaceutical industry for products that require solubilization in hot water before consumption, such as vitamin cocktails, mixtures of ingredients to treat colds, dizziness, headaches or as delivery systems for antireflux drugs [7]. These ecofriendly products reduce the amount of non-biodegradable waste materials, thereby minimizing the negative impact on environmental pollution [8,9]. From its vast use in food processing and biotechnology in a constantly growing market of approximately 10 billion USD by 2021, the use of alginate’s attractive properties have expanded to the biomedical and pharmaceutical industries. To this end, its biocompatibility, biodegrability, high capacity to incorporate and release proteins, cell affinity, strong bioadhesion and absorption characteristics have led to the development of “smart” polymers used in numerous medical applications ranging from wound healing and tissue regeneration to drug delivery agents in the management of several pathological conditions including obesity and diabetes [10,11,12]. For this latter use, numerous delivery systems have been developed by using alginate, such as hydrogels, tablets, capsules, liposomes, nanoparticles, beads, microspheres and others. Their development requires innovative extraction technologies such as those used in developing macroalgal hydrocolloids, like enzyme-, microwave-, ultrasound, or fluid-assisted processes [13,14]. Although much is known about the sources, methods of extraction, properties, functions and applications of alginate, its physiological characteristics, modulation of the body’s physiological functions and safety have led to an increasing interest in improving its efficacy and in the development of new applications for the pharmaceutical industry and biotechnology.

Several review articles have focused on recent technologies, developed for the specific biomedical applications of various formulations of alginate and designed for a controlled release and drug and protein delivery ranging from engineered bioresorbable functional materials to cell-based therapies and three-dimensional cell cultures and bioinks, to name a few [15,16,17]. Therefore, this review summarizes our current understanding of the use of alginate, from the food industry as a food additive or component of packaging materials to advanced applications in the medical and pharmaceutical industries. In the process, the paper highlights advantages and disadvantages of using alginate in biomedical applications, processing techniques, the combinations of compounds and preparations incorporated with alginate and their effects on several in vitro and in vivo applications. Considering the increased prevalence of metabolic disorders worldwide, such as obesity and diabetes, conditions that are associated with high costs and no immediate cure in sight, this review further highlights the most recent evidence on the use of alginate in the management of obesity and diabetes, from its potential role in controlling food intake, energy balance and glucose homeostasis to serving as a microcarrier for drug delivery in the control of these pathologies. This includes the most recent findings on using alginate in the delivery of probiotics, synbiotics and other gut microbiota composition modulators, a topic of heightened research interest, given the implication of gut microbiome in health and disease. Finally, the perspective section discusses current gaps in the use of alginate products in the food and biomedical industries, offering future guidance and directions for enhancing product development and understanding the associated underlying biological mechanisms.

## 2. Alginates: Properties and Challenges

Alginates are polysaccharides composed of β-D-mannuronic (M) and α-L-guluronic (G) acid units that form regions of M-blocks and G-blocks as well as blocks of alternating sequences (MG blocks). This structural organization depends on alginates’ sources. For example, leaves of *Laminaria hyperborea* contain a high amount of mannuronic acid, unlike the stipe and outer cortex, which contain a high amount of guluronic acid. Similarly, fruiting bodies of *Ascophyllum nodosum* contain a higher amount of mannuronic acid, compared to old tissue, which is abundant in guluronic acid [18]. Sodium alginate is readily available, is ecofriendly and is relatively cheap to produce. This, together with several other advantages such as biocompatibility, biocompostability and nontoxicity, has led to the development of multiple alginate-based applications in the food and biomedical industries [19] (Figure 1). Sodium alginate represents alginate’s most common salt.

However, due to its characteristics, alginate also has a number of limitations, such as a poor stability or low mechanical and barrier properties, incompatibility with heavy metals and heat treatment instability, some of which cannot be corrected later (Figure 2). These properties can be improved by combining alginate with other biopolymers, especially protein-based ones, or with synthetic polymers, by modifying or combining the treatments applied during production.

Although it is possible to obtain alginate from both algae (mainly Laminaria hyperborean, Ascophyllum nodosum, Macrocystis pyrifera, and to a lesser extent Laminaria digitate, Laminaria japonica, Lesonia negrescens, Sargassum sp., Eclonia maxima) and bacterial sources (mucoid strains of Pseudomonas aeruginosa or Azotobacter vinelandii), alginates currently used for commercial purposes come only from algae. The special feature of bacterial alginates is that they present O-acetyl groups, which are not available in the structure of algal alginates and possess higher molecular weights compared to the algal polymers [20]. The sequence and copolymer composition or molecular weights vary depending on the source and species that are involved in the production of the copolymer. Due to the abundance of algae in the marine field, there is a large amount of alginate material present in nature [21]. Industrial alginate production is at approximately 30,000 tons per year, of which approximately less than 10% is biosynthesized [22]. Therefore, the design potential of sustainable alginate-based biomaterials is very high.

Alginate is a biodegradable and biocompatible natural material able to absorb 200–300 times more water than its own weight [23]. As such, alginate has become an extremely important family of polysaccharides with a wide range of uses and applications in a variety of fields. According to the European Food Safety Authority, alginic acid and its salts are authorized for use in *quantum satis* in a wide range of foods, including foods for infants and young children with special medical purposes. The evidence reviewed shows that alginic acid and its salts are practically undigested, not absorbed intactly, and that they are partially fermented by intestinal microbiota in humans. Toxicity data in rodents showed no adverse effects at the highest tested dose of 13.5 mg sodium alginate/kg/day in a 90-day study when given subchronically. There were no concerns reported with respect to the genotoxicity of alginic acid and its salts [24].

Alginates have been shown to act on human macrophages by inactivating the proinflammatory cascade, leading to resolving inflammation characteristics in the healing process. In their study, Yang & Jones, [25] showed that sodium alginate causes natural immune responses through the activation of the innate-associated transcription factor NF-kappaB that controls the transcription of DNA, cytokine production and cell survival, a mechanistic pathway similar to that observed in pathogen recognition. Alginate oligomers with defined chemical structures have cytokine-inducing activities in a structure-dependent manner. For example, enzymatically depolymerized alginate oligomers induced the secretion of cytotoxic cytokine from human mononuclear cells [26]. Furthermore, highly purified sets of alginate oligomers are interesting agents, not only as potential therapeutic agents but also as a tool for analyzing the pattern recognition mechanism [27].

Alginate is considered an important source of dietary fiber, with studies showing its positive effects in reducing cholesterol and glucose uptake, and with benefits in cardiovascular and gastrointestinal diseases [28]. Sodium alginate is also widely used in the pharmaceutical industry as a coating element of drugs, facilitating the controlled release of active substances, maintaining flavor compounds and allowing the incorporation of various ingredients. In addition to the other benefits, all these properties contribute to improving the quality of drugs. Before food or medicine is ingested, the perception of flavor during oral processing depends on the rate of release of taste and aroma compounds from the salivary bolus as they bind to their respective receptors in the oral cavity. A reduction in the rate of release of flavor or any bioactive compounds activating taste and olfactory receptors results in a loss of aroma and taste perception as food passes through the mouth. In order to overcome this, a common strategy is to increase the flavor concentration and taste compounds in foods. However, this leads to unhealthy products with excessive levels of sugar, flavorings and sodium.

The development of safe, healthy and effective mucoadhesive biopolymers that improve residence time and promote the delivery of flavor compounds in the oral cavity has been studied. For example, researchers have set up a faster and more sustainable biomimetic alternative in order to evaluate the retention of aromatic compounds in the presence of different polymeric mucoadhesives, for instance through the encapsulation of different concentrations of mucin in alginate calcium spheres [29].

In addition to its well-known uses in the food, medical and pharmaceutical industries, alginate has also been used for wastewater treatment, representing a high-quality entrapping agent for immobilizing nitrifying microbes and removing ammonia nitrogen. Finally, alginate presents a high water solubility, good mechanical properties, chemical stability, good resistance to biodegradation, little toxicity to organisms, little loss of bacteria activity and low costs [30].

## 3. Sodium Alginate in the Food Industry

Recently, edible films and coatings have become increasingly used due to their similar characteristics to conventional, synthetic materials. An edible film represents a thin-layered biopolymer structure that can be consumed and that is usually applied onto the surface of food products by casting, coating, spraying, dipping, extrusion or brushing [31].

The most common technique to develop food films is the casting method. This method uses a mixture of alginates with deionized water, plasticizer and other ingredients, followed by hot-plate magnetic stirring and pouring on a drying surface. The surface used for drying plays an important role in the final physical characteristics of the film. Thus, silicone foils have become more and more often used, due to their low adhesiveness and their ability to form fine films. Glass substrates or Teflon foils can also be used, but they have a high adhesion, and, in the absence of an optimal plasticizer content, the obtained films cannot be detached from the foil used for drying. Coating is applied to products’ surface by wrapping them in a film-forming solution, and it is usually used for packing fruits and vegetables. Spraying, dipping or brushing represent direct coatings on the surface of food (usually vegetables, fruits and meat) and indirect coatings (on the surface of packaging materials) and require the use of tools, such as brushes or spatulas [32]. Spraying is recommended when the coating is applied on only one side of a product.

Edible coatings represent a thin layer on the product surface. Due to the fact that many studies about coating applications have been conducted on a small food laboratory scale, commercial applications still face certain limitations. In order to implement a recycling process that does not waste too much coating solution and decreases the microbial load of the solution during recycling, coating application methods can be readjusted. Thus, by designing a spraying method for irregular surfaces or by designing industrial-sized vacuum tanks, the disadvantages regarding their method of application can be prevented [20].

These films and coatings are considered a barrier in order to prevent the loss of foods’ volatile flavor compounds and to protect the product from microorganisms’ damage. Since they can be easily incorporated with natural substances (antioxidants, additives, preservatives, dyes), alginate films and nanocapsules help to improve the product’s appearance and make it more attractive for a consumer [33]. Furthermore, alginate can be used to preserve the qualities and prolong the shelf life of foods, especially for the controlled release of incorporated compounds. Table 1 depicts the ability of alginate to serve as an incorporating material for various natural substances, such as EOs (essential oils), natural extracts, fruit and vegetable purees, or vitamins, and it shows their effects on foods or packaging.

Thus, alginate-based films can reduce spoilage, food waste and food illness outbreaks or food recalls [51]. Although these characteristics show the importance of alginate primarily in food applications, they also have relevance for pharmaceuticals, cosmetics and other consumer good products.

## 4. Applications of Alginate in the Medical Field

Recently, biopolymer films have been intensively studied, especially due to their biocompatibility and biodegradability properties, and have been used in wound healing, tissue engineering [51] and drug delivery [52] applications.

### 4.1. In Vivo Applications

Alginate and alginate-based products have been extensively tested and evaluated using various animal models of human disease, such as tumors, skin inflammations or atopic dermatitis, diabetes, osteoporosis and others, which mimic the physiological environment in order to reliably predict the biocompound behavior, its safety and their effects (Table 2).

### 4.2. Processing Techniques and Applications of Alginate in the Medical Field

Due to the multiple applications of alginate in medicine, numerous processing methodologies have been developed and implemented. Depending on its subsequent use, alginate can be processed into microbeads, dressings, foams, and hydrogels or aerogels.

#### 4.2.1. Microbeads/Microspheres

Alginate microbeads are obtained through extrusion, spray-drying, and single or double emulsion.

##### Extrusion

This is known as the ionic gelation method; the gelation is highly dependent on the concentration of CaCl_2_. For the production of microbeads, the suspension containing the active substances is suspended and vigorously mixed. Using a syringe, the mix is suspended into a calcium chloride solution with mild agitation for one hour in order to cure the calcium alginate gel beads [75]. The beads are collected and dried in a vacuum for approximately 12 h, after which they can be stored in polyethylene bags or tightly closed glass bottles at room temperature before use. Because applications for pharmaceuticals or bioreactors require large quantities, other techniques have been used for a larger scale production. For example, atomization is the transformation of a liquid into droplets that are solidified in order to form individual particles. This technique permits a production that reaches up to tons per day. The droplet size can be adjusted according to preferences and subsequent uses by replacing the accessories of the atomizer (a rotating disk with a needle system).

##### Spray-Drying

Seen as a reliable and reproducible solution, spray-drying represents a single-step process that converts a suspension, a dispersion, an emulsion or a liquid into fine particles. With a wide use in the pharmaceutical industry and research, the spray-drying principle involves four steps: *(i)* atomization of liquid and transformation into very small drops, *(ii)* injection into a drying chamber containing hot air (or an inert gas, such as nitrogen), *(iii)* instant drying of atomized droplets into solid particles, *(iv)* collection of solid particles into a drying chamber [76].

The size of droplets and the configuration of the equipment depends on the type of atomizer and the properties of the fluid. There are several advantages in using this method, such as solid droplets having a physical and chemical stability, the drying of fluid into solid droplets being a quick process, the method being versatile (suitable for multiple types of emulsions, solutions and suspensions), as well as the process being easily reproducible, controllable and tailorable, and cheaper than other drying methods (such as freeze-drying, which involves a cooling step). The disadvantages refer to the cost of the industrial equipment, which requires periodical cleaning, maintenance and optimization, or the waste of energy and heat during the operation. Spray drying is applied in drug delivery to improve the solubility, wettability (induced by the use of a hydrophilic polymer) and crystallinity of the drug, bringing it into a more favorable state.

##### Emulsification

For this process, the experimental set-up is easy and relatively cheap. Generally, the emulsification technique involves two steps: the formation of polymer droplets into an emulsion system and the process of drying/hardening the drops. One characteristic of this technique is that the drugs are dispersed in an alginate solution before emulsion. The emulsion is obtained by adding the mixture in an oily phase that contains the suitable surfactant. CaCl_2_ is used as a chemical cross-linking agent and is mixed into the emulsion. The microspheres are washed (to remove the oil phase), collected and dried at room temperature [77]. Liquid paraffin is the widest oil phase used, although other types of oils can be used, such as soybean oil, olive oil or sunflower oil. The choice of the type of oil influences the final product because the size and uniformity of the microspheres depend on its viscosity. The role of the surfactant is to stabilize the emulsion by lowering the interfacial tension between hydrophilic and hydrophobic molecules and to stabilize emulsion droplets against coalescence [78]. The size of the microspheres developed through the emulsification technique is smaller than those produced from the extrusion method.

#### 4.2.2. Wound Dressings

Usually, wound dressings are synthetized from synthetic polymers and biopolymers. When used as a wound dressing, sodium alginate is usually combined with calcium chloride, resulting in pads or ropes. Ca^2+^ from the dressing interacts with Na^+^ in the fluid from the wound, so that the dressing fiber swells and is partially transformed into a gel that moisturizes the wound bed and accelerates the healing process. The process for the development of alginate wound dressings involves a series of steps: *(i)* a mixture of sodium and calcium alginate is coated with ethyl alcohol to prevent the gelling of the alginate upon contact with water; *(ii)* addition of deionized or distilled water and obtaining the alginate solution, *(iii)* impregnation, followed by drying of a woven or nonwoven material with the alginate solution, *(iv)* mechanical softening of the dressing to obtain a soft and flexible material.

In addition to the ability to form dressings, alginate can incorporate various natural substances that, when released together with calcium ions, can activate prothrombin and improve hemostasis.

A good product for wound care must meet a series of qualities: (a) to have antimicrobial effects (antiviral, bacteriostatic or fungistatic), (b) to be breathable, (c) to be nontoxic and nonallergenic, (d) to be hemostatic, (e) to be biocompatible, (f) to be able to perform and display mechanical resistance, (g) to facilitate incorporating drugs. For example, alginate wound dressings absorb a large quantity of liquid into the structure of the fiber, in addition to those from the textile structure fibers. Alginate wound dressings have antimicrobial and hemostatic properties, thus promoting wound healing. These characteristics are useful, given that infections lead to delays in the healing process [79]. They are also widely used in the management of highly exuding wounds such as surgical wounds, leg ulcers and pressure sores [80].

Currently, solutions are being sought to replace synthetic polymers, which are also widely used in medicine due to their advantage of being inert, having a high mechanical performance and the fact that, unlike biopolymers, they can be easily handled and processed. In order to match the rate of tissue regeneration, the properties of natural polymers require optimization, by establishing mechanical strength and a rate of degradation [81]. Wound healing involves the regeneration of dermal and epidermal tissues through consecutive stages of inflammation, migration, proliferation and maturation [82]. This process can be complicated by sepsis, disruption of the tissue and skin layer, maggots’ formation, and the extension of infection to adjacent and interior organs [83]. To prevent against infectious organisms, wound dressings loaded with antimicrobial agents or healing agents have been developed [84]. For example, Üstündag˘ Okur et al. [85] developed sodium alginate, carbopol and chitosan films for Mupirocin dermal delivery. In vitro drug release studies demonstrated that Mupirocin release was increased and that, in most cases, the drug was solubilized. Ex vivo permeation studies also showed that the film could be safely applied for contact delivery because a very low amount of Mupirocin could cross the epidermis layer, resulting in the speeding up of the regeneration of the epidermal layer. Furthermore, compared to the commercial product, the film offers better advantages for epithelialization, granulation tissue thickness and angiogenesis. Considering the challenges of the new century, these natural materials have begun to replace synthetic polymers, ceramics and metal alloys [86]. Biomaterials are defined as materials intended to interface with biological systems in order to evaluate, treat, augment or replace any tissue, organ or function of the body [87]. To be used in the medical industry, biomaterials must be inert and not interact with the host organism. Thus, natural materials have been extensively used for this purpose, and the applications to replace the affected tissues have been in use and reported for a long time in the specialized literature.

#### 4.2.3. Foam Dressings

Currently, wound dressings have been widely replaced with foam dressings. The foam dressing provides a better protective cushioning effect over the wound because it is thicker than the wound dressing. They not only prevent exudate pooling but also maintain hydration so that the appropriate moisture environment promotes epithelialization and healing through encouragement of cell migration [88]. Alginate foam dressings are made by vigorously homogenizing a mixture of an aqueous solution of water-soluble sodium alginate with a sequestering agent, a plasticizer and an active surface agent, followed by the addition of a di- or trivalent metal ion to form water-insoluble hydrogel. The resultant mixture is poured into a tray and placed in the freezer to obtain a frozen alginate hydrogel. The final stage involves lyophilizing the mixture in order to eliminate moisture.

Alginate foam dressing can be enriched with active agents, such as silver, used as an antimicrobial agent, and asiaticoside, a herbal wound healing agent. In this preparation, alginate generates the highest silver and asiaticoside release, improves the foam characteristics with an increase in the absorption property and compressive strength, has a comparable antimicrobial effect in the disk diffusion test and shows noncytotoxicity [89]. Finally, the alginate dressing preserves a moist environment around the wound, recovers the wound tissue to a normal state, while ensuring the disappearance of granulation tissue [90] and eliminating the possibility of damaging the tissue when the wound is removed [91].

#### 4.2.4. Hydrogel Dressings

Sodium alginate has been used to develop hydrogel dressings. Due to their capacity to produce an ideal hydration environment for healing, hydrogel dressings are of great interest [92]. To obtain hydrogel dressings, sodium alginate, deionized water and H_2_S are mixed and vortexed for one hour. CaCl_2_ is added and gently mixed for the alginate polymer chains’ ionic cross-linking initiation [93]. Other cations, such as divalent cations (Ca^2+^, Sr^2+^, Cd^2+^, Zn^2+^, Ba^2+^, Cu^2+^) or trivalent cations (Al^3+^, Fe^3+^) can lead to the formation of ionically cross-linked alginate hydrogels. The association between divalent cations and the alginate chain is described using the “egg-box” model (external gelation, preferred for the synthesis of materials applied in bone tissue engineering) and gelling mechanism (internal gelation, preferred for in situ hydrogels, suitable for injectable alginate applications and considered to be the most common and easiest way to encapsulate soluble and insoluble drugs). Alginate hydrogels can be chemically modified by the alteration and modification of the molecular weight (alginates with a higher molecular weight produce stiffer gels, despite those with a medium and low molecular weight allowing a greater degradability and cell proliferation).

The physiological soft tissue is reassembled because it presents a low interfacial tension, oxygen permeability, and good mechanical and moisturizing properties [94]. For these reasons, polysaccharides that display hydrogel-forming properties are considered to be advantageous as a wound dressing material [95,96]. With 3D cross-linked polymeric networks, a high biocompatibility and biodegradability, hydrogels are applied effectively as controlled drug delivery and wound healing systems. In order to be effective, hydrogels must provide adhesiveness, elasticity and durability, and should be occlusive and impermeable to bacteria. Alginate is capable of forming hydrogels under very mild conditions at 25 °C [97]. However, no adhesive alginate hydrogels support the growth of pluripotent stem cell-derived intestinal organoids due to the fact that alginate properties that support human intestinal organoids’ (HIOs) growth in vitro lead to HIO epithelial differentiation when transplanted in vivo [98]. To improve their properties, hydrogel films can be incorporated with different natural substances. For example, alginate/*Aloe vera* hydrogels have been developed for wound healing applications, thus combining the haemostatic properties of calcium alginate gels with the therapeutic properties of *Aloe vera.* The results suggest that alginate/*Aloe vera* hydrogel films can be explored as wound dressing for dry and exuding wounds [99].

#### 4.2.5. Alginate Bioaerogels

Bioaerogels are another novel class of porous materials that represent a group of structures with certain characteristics, such as a low density, high porosity, thermal resistance, ultralight weight and low dielectric constant, which make them great candidates for a wide range of applications, such as thermal insulation, energy storage devices, environmental clean-up, medical and pharmaceutical uses, food industry and others [100,101,102]. The first step in obtaining the aerogel is the formation of a gel from an aqueous solution, with the help of a crosslinking promoter that can be physical (such as pH, temperature) or chemical (e.g., crosslinking compound). The alcogel is formed by replacing the water from the gel structure with a solvent (alcohol, usually ethanol). Alcohol is extracted from the gel by supercritical carbon dioxide (scCO_2_)-assisted drying.

These solid materials are developed by replacing the solvent trapped in the gel structure with a gas (e.g., air), leading to an open-ended mesoporous structure typically composed of 95–99% empty space [103]. Polysaccharide-based aerogels obtained with polymers with natural resources (alginate, agar, starch, cellulose, chitosan, chitin or pectin) are sustainable materials that can substitute silica aerogels in biomedical and pharmaceutical applications [104,105,106]. Different medical applications, such as tissue engineering, drug delivery or biosensing have been developed due to the advancement in bioaerologels research. As such, aerogels can mimic extracellular matrices in the body [107] and allow high drug loads due to their good biocompatibility, high surface/volume ratios and large interior surface areas. To exploit these properties, different techniques have been developed for loading drug molecules into aerogels before gelation, through the aging step, or during the adsorption or precipitation process [91].

The structure, composition or hydrophobic nature of biopolymer aerogels influence their features, such as the dispersion of the drug in the aerogel matrix, the crystalline/amorphous phases of the drug, and the accessibility of the loaded drug for the solvent through the release process. The drug release profile of aerogels is also determined by a combination of these properties [108]. In fact, the volume and surface area of the aerogel are the key parameters controlling the release of the drug [109]. To take advantage of this property, 3D multimembrane alginate-based aerogels with onion-like architectures have been developed, forming free spaces perfectly suited for cell or drug incorporation that increase drug loading and prolong drug release [110]. Among the multiple types of aerogel, aerogel microspheres have shown excellent properties for biomedical applications such as tissue engineering [111]. The data thus far show that bioaerogels are recognized as promising biomaterials that need to be further studied in order to develop new biomedical applications.

### 4.3. Alginate and Tissue Engineering

Alginate gels have been widely used as delivery vehicles for various angiogenic molecules. This property is extremely important and represents a promising alternative when treating patients suffering from a restricted or obstructed blood flow caused by coronary and peripheral arterial diseases [86]. Alginate gels have been extensively studied for tissue engineering applications as cell encapsulation material as well as for in vivo cell delivery as an injectable 3D matrix. Several studies demonstrated that calcium alginate gel, although easy to obtain, exhibited poor biodegradation, bioresorbability and cell adhesion [112]. For example, ultrapurified alginate gels are biocompatible with human lumbar intervertebral disc cells and promote extracellular matrix production after discectomy. This demonstrates sufficient biomechanical characteristics without material protrusion, making them safe to use and efficacious. In fact, ultrapurified alginate gels represent a novel therapeutic strategy after discectomy in cases of lumbar intervertebral disc herniation [113]. Because alginates gel can be introduced into the body in a minimally invasive manner or to fill irregularly shaped defects, they have the advantage of participating in bone and cartilage regeneration. Because of their adaptability to chemical changes with adhesion ligands and the controlled release of tissue induction factors such as growth factors and bone morphogenetic proteins, alginate hydrogels represent a viable alternative for the regeneration of articular cartilage [111].

As mentioned above, sodium alginate has been extensively used as a biomaterial, due to its biocompatibility and hydrophilic properties, and, at the same time, it is easy to inject [113]. This allows its administration by minimally invasive surgical techniques [114]. The hydrogel represents an off-the-shelf solution that may be applied to any irregular defect without prior preparation. The nature of the material allows its reshaping during application to match the specific structural needs, finally resulting in the formation of a shear-resistant and integrated layer with complete topographical matching to surrounding defects and the opposing articulating surface [115]. Thus, hydrogels can be manufactured in a variety of different shapes, such as films, discs, rods and microparticles, and are used in a variety of applications [116].

### 4.4. Alginate as Drug Delivery Vehicle

The novelty of applications of nanoscience and nanotechnology in healthcare led to the genesis of a field called nanomedicine [117]. At present, nanomedicine uses nanomaterials such as nanoshell, nanobiosensors, nanovaccines, nanorobots and nanocapsules for various biomedical applications [118]. Nanoparticles are also being investigated and used in many areas of medicine for specific drug delivery, for lowering the dosing of active agents in combination therapy with minimum side effects, and to harness more potent drugs, which cannot be clinically utilized by conventional drug delivery. In response to the growing threat of microbial drug resistance, nanotechnology has become a major area of interest because of its many unique characteristics [119]. These include the utilization of materials that, at the nanoscale, have inherent antimicrobial properties and the incorporation of known therapeutics into nanovehicles to enhance delivery and improve efficacy.

The most commonly used method of drug delivery has always been through the oral route. Prolonged-release drugs, administered orally, face two challenges: a short gastric residence time and an unpredictable rate of gastric emptying [120]. The stomach anatomy and physiology must be taken into account in the development of gastro-retentive dosage drug forms [121]. To pass through the pyloric sphincter into the small intestine, the particle size should be in the range of 1 to 2 mm [122]. Factors that control the gastric retention of dosage forms and that should be considered include: *(i)* the density of dosage forms (a density lower than the gastric contents can float to the surface, while high-density systems sink to the bottom of the stomach); *(ii)* the shape and size of the dosage form (ring-shaped and tetrahedron-shaped drugs have a better gastric residence time); *(iii)* food intake and its nature (the gastric retention of drugs depends on the volume and viscosity of the food intake); *(iv)* the caloric value and feeding frequency; *(v)* the gender or posture (females have slower gastric emptying rates than males); *(vi)* the age (gastric emptying is slower in the elderly), *(vii)* diseases (e.g., gastric emptying is slower for diabetics) [123]. Because biopolymers have mucoadhesive properties, they can be used to extend the contact time of products with the mucous layer of the gastrointestinal tract [124]. This property is widely exploited in the pharmaceutical industry in order to prolong drug resistance and to retain the taste or aroma molecules in the oral cavity [125,126]. The mucoadhesives can be part of tablets, sprays, films, patches or other pharmaceuticals. Since aroma release is controlled by particle size, the encapsulation of aromas has been a frequent topic of research [127,128].

Several methods have been employed to use alginate in drug delivery. In their study, Cheng et al. generated two types of ketoprofen delivery using calcium and chitosan-alginate beads. The release profile studies indicated that, in both preparations, ketoprofen had a slow and sustained release in the gastrointestinal tract, relieved immediate pain, and reduced administration time and gastrointestinal irritation. These findings demonstrate that using alginate for drug encapsulation is an effective tool for short- and long-term applications [129]. Indeed, alginate nanoparticles have been used for cancer drug delivery (e.g., doxorubicin, 5-Fluorouracil or methylene blue) and as an antibiotic and antimicrobial drug delivery system for amoxicillin, ciprofloxacin, streptomycin, nisin, rifampicin, isoniazid, pyrazinamide and ethambutol [130].

Alginate-based nanoparticles for drug delivery can be prepared using various methods, including: *(i)* ionic cross-linking (ionotropic gelation). This preparation is simple and mild, and is produced by alginate cross-linking with Ca^2+^, Ba^2+^ and Al^3+^; *(ii)* emulsions, where the size of nanoparticles is usually below 250 nm, which is the desirable size for drug delivery applications due to an enhanced cellular uptake; *(iii)* polyelectrolyte complexation. This results from mixing oppositely charged polyelectrolytes. Other factors, such as pH, temperature and stirring speed, may play major roles in controlling the size of alginate nanoparticles [130]. Of critical importance is the fact that alginate-based nanoparticles avoid aggregation during blood circulation by reducing the interaction with serum proteins and usually do not concentrate in internal organs while they deliver drugs or proteins [131].

### 4.5. Alginate in Probiotic Encapsulation

Probiotics are live microorganisms (bacteria or yeasts), which when ingested or locally applied at a sufficient concentration confer health benefits for the host [132]. A microorganism is considered probiotic if it is a normal resident of the gastrointestinal tract, survives the passage through the stomach, and maintains its viability and activity in the intestine [133]. On the other hand, prebiotics are nondigestible substances, resistant to hydrolysis in the stomach and small intestine, and are therefore included in the category of dietary fibers [134]. They play an important role in the nutritional, physiological and immunological processes, constituting a viable alternative for improving probiotic activity. A major challenge to the food industry remains the preservation of a microorganism’s viability throughout the product shelf life, since certain bacteria are extremely sensitive to environmental factors such as acid and the presence of oxygen [135,136]. Although the long-term efficacy of probiotics in the prevention or treatment of certain diseases in humans is still debatable, probiotics have been proven to be safe for human use. Nevertheless, several studies suggest that, when used in combination, some bacterial strains act in a synergistic fashion to improve disease outcomes, including those associated with metabolic disorders such as obesity and diabetes [137].

Maintaining probiotics’ cell viability is critical, since they can be denatured by the low gastric pH, bile acids and enzymes in the small intestine or an antibiotic treatment. In order to improve the viability and stability of probiotics, different techniques have been employed. Probiotics can be incorporated into food matrices through edible films that confer protection from the damage induced by environmental conditions or by food processing, manipulation and storage [138,139]. Encapsulation involves the coating or entrapment of one substance (active agent) into another substance [140]. The encapsulating substance is called coating, membrane, shell, capsule, carrier material, external phase or matrix [141]. Encapsulation has been widely used in various domains, including pharmaceutics, medicine, food, agriculture and biotechnology. Vitamins, minerals, enzymes, antioxidants, colorants, amino acids and sweeteners have been successfully encapsulated in the pharmaceutical and food industries [142].

Several methodologies have been used when microencapsulating probiotics. These include: spray-drying, which uses water-soluble coating materials; spray congealing, which uses waxes, fatty acids, soluble and water-insoluble polymers, and other monomers as coating material; fluidized bed coating/air suspension, which utilizes soluble and water-insoluble polymers, lipids and waxes as a coating material; and coacervation or phase separation technique, which uses encapsulating materials as water soluble polymers [143]. Because the spray-drying process produces powders with micrometer-scale particle sizes, this allows the addition of probiotics to a wider range of foods. This procedure results in nanoparticles that confer a smoother mouth-feel texture than microbeads without changes in the sensory characteristics of the food, with a positive impact on food intake [144]. However, spray-drying is limited by the high viscosity of the alginate, and in some studies it actually decreased cell viability [145]. Encapsulation with sodium alginate has been shown to improve bacterial availability during the spray-drying process [146]. Furthermore, sodium alginate has been used in combination with other compounds such as succinic anhydrate, which not only has protective metabolic effects in obesity but also improves bacterial cell viability [147].

Alginate has been used for the encapsulation of several strains of probiotics and tested during exposure to various adverse environmental conditions. In addition, prebiotics have been used to enhance protection for probiotics due to the symbiotic relationship between alginate and prebiotics resulting in 3D networks of microcrystals that interact together and form small aggregates [148]. The main findings highlighting the advantages of the microencapsulation of several bacterial strains in alginate-based composites, resulting in increased viability and storage stability, are presented in Table 3.

The microencapsulation of probiotics improves the storage time of viable bacteria at room temperature and allows the incorporation of probiotics into a wide range of food products without the risk of degradation of sensory characteristics [160]. Extrusion and emulsification are the most common methods used to develop sodium alginate microcapsules. Compared to the extrusion method, emulsification is milder and simpler. The emulsified microcapsules contain a small particle size and do not affect the taste of the products in which the microcapsules are added. In addition, emulsification is more suitable for large-scale production [149].

The development and success of a new encapsulated product is dependent on the consumer’s behavior and attitude, such as perception, preference, acceptance and choice of the product [161]. Factors that affect consumers’ choices may be: (a) biological (e.g., palatability, hunger, satiety); (b) economic (e.g., cost, income, education, knowledge level); (c) physical (e.g., time or skills); (d) meal attributes (e.g., convenience, familiarity); (e) social determinants (e.g., culture, relatives, family norms); (f) physiological determinants (e.g., mood, stress, history influences); and (g) other (belief, optimistic bias, restrictions) [162].

### 4.6. Alginate in the Management of Diabetes

Diabetes mellitus is a complex disease affecting millions of people worldwide. The World Health Organization (WHO) predicts that by 2030, diabetes will be the seventh leading cause of death. Diabetes is a major cause of blindness, kidney failure, heart attacks, stroke and lower limb amputation [163]. Insulin is one of the main treatments for diabetes, but long-term injections can lead to a lack of patients’ compliance and pain. An alternative that has attracted the attention of the medical and pharmaceutical industries for some time is the oral administration of insulin. However, this has been met with challenges, as enzymatic hydrolysis and the membrane barrier reduce insulin’s absorption efficiency. In order to overcome these barriers, several noninjection carrier systems have been developed to improve the permeability and delivery of protein drugs such as insulin through the oral route [164]. Polymers have been used to develop devices for insulin oral delivery. In vivo model studies have shown the efficacy of sodium alginate and chitosan nanoparticles when tested as oral delivery insulin carriers. The mixture of alginate with chitosan improved blood glucose levels and insulin bioavailability by approximately 8.11%. Several alginate analogs have been used to enhance biocompatibility. These analogs do not trigger strong inflammatory responses and possess similar behavioral characteristics under similar conditions [165]. For example, chitosan and alginate can tolerate acidic pH and do not show toxic effects during in vitro and in vivo studies [166].

No systemic toxicity was found after its perioral treatment; therefore, nanoparticles are a promising device for potential oral insulin delivery [167]. Their small particle size, sustained release, good stability and strong absorption capacity make nanoparticles a viable and beneficial alternative for use as insulin carriers. The nanoparticles’ fast dissolution and high adsorption capacity for surface loading is due to their large surface area under the same volume conditions. The same features provide protection from proteolytic enzymes, enhanced mucoadhesion and increased retention. Because they are resistant to the acidic environment and are slowly released in the gastrointestinal tract, alginates have also been used as coating material for the oral administration of antidiabetic drugs [163]. These findings show that alginate is a good candidate for oral drug delivery that requires sustained and prolonged delivery to counteract fluctuations in blood glucose levels, as is the case with hypoglycemic drugs (Figure 3).

### 4.7. Alginate in the Management of Obesity

The World Health Organization (WHO) has long recognized obesity as a global epidemic, and recent data shows that its incidence has tripled over the past 30 years. Moreover, 1.9 billion adults and 340 million children and adolescents worldwide were overweight or obese, according to 2016 statistics [164]. Obesity is a complex multifactorial disease resulting from genetic, epigenetic, physiological, behavioral, sociocultural and environmental factors. Consequently, management of obesity requires a comprehensive approach, among which dietary modifications and physical activity play the most important role. Observational as well as interventional studies show that the increased consumption of dietary fiber is associated with weight loss, and alternative treatments with dietary fiber have made their way up to clinical trials, with various degrees of success [165]. Given its effects in reducing inflammation and postprandial blood glucose, alginate has been considered as an agent in the treatment of obesity. As such, sodium alginate is marketed as a weight loss supplement, either in standalone formulations or when incorporated into various foods. Capsules containing sodium alginate dissolve in the stomach and expand into a soft, gel-like solid, which remains stable in the gastric acid environment and maintains satiety for 6 to 8 h, after which it dissolves due to the neutral or alkaline environment in the small intestine [166]. In addition to increasing satiety, alginate inhibits gastrointestinal enzymes, reduces cholesterol and glucose uptake, and controls lipid digestion. However, when used as an additive, in the concentrations commonly used in foods (especially as a stabilizer, E401), alginate has not been shown to have such effects [167]. Several food matrices, such as bread, cereal bars or beverages, have been used as targets for alginate incorporation [168]. When alginate was incorporated in bread, it maintained its inhibitory properties, despite cooking and digestion [169]. Although there were no significant differences as the temperature increased from 37 to 100 and 150 °C, alginate lost 88% of its inhibitory properties after exposure at 200 °C. Therefore, the process of baking bread did not reduce the molecular size of alginate and did not affect its inhibitory properties.

Alginate can prevent the action of pepsin without affecting trypsin due to the pH-dependent interaction between alginate and protein. The binding of alginate to pepsin at low pH and the subsequent formation of a precipitate removes the enzyme from the solution and, thus, makes pepsin unavailable to the substrate [170]. Alginates also have the ability to interact with the substrate and not only with the enzyme. For example, alginates with a high G block content are known to interact with glycoproteins, to increase fatty acid excretion and to improve weight management.

The physicochemical properties of alginates are dependent on the sequence of the M and G residues. As such, the ratio of mannuronic and guluronic acid residues depends on the biological source and maturation state of alginate seaweeds [171]. The ability of alginate to inhibit the action of pancreatic lipase also depends on the same chemical structure, that is to say, respectively, on the G and M block content. Reducing pancreatic lipase levels leads to triacylglycerol breakdown, resulting in lower amounts of this molecule being absorbed by the body. Alginates with a high G block content have a higher inhibitory capacity for lipase than those with a high M block content. Thus, the activity of pancreatic lipase can be controlled depending on the alginates used [172]. The substrate is also important. For example, when a synthetic substrate was used, alginate inhibited pancreatic lipase by 72.2 ± 4.1% vs. 58 ± 9.7% when olive oil was used as a natural substrate [173].

Another mechanism by which alginate acts to control obesity is by reducing the uptake of cholesterol and glucose. In a pilot study conducted by the University of Sheffield (UK), it was shown that alginate reduced the absorption of cholesterol and glucose in overweight male subjects. Similarly, when alginate was incorporated in bread it resulted in a significant reduction in lipid absorption and lowered triglyceride concentrations [174]. These studies demonstrate the effectiveness of alginate in reducing the caloric intake and subsequent body weight [175] (Figure 3). An important factor in the development of obesity is excess caloric intake. Several studies showed that alginate reduced satiety feelings, gastric emptying and energy intake [176]. *Maaljaars* et al. gave 20 healthy subjects alginate, at two different doses, (9.9 g and 15 g), as a liquid preload prior to meals. They found that the higher dose of alginate increased satiety, decreased hunger and food intake, and increased the tendency to feel full. This may be due to an increase in the viscosity of the intestinal contents with the formation of alginate gel. The water layer is thickened, and soluble fibers may slow the rate of nutrient absorption. This effect could delay the nutrients’ transit in the intestinal lumen. This increased nutrient exposure results in the release of inhibitory gut peptides, such as glucagon-like peptide-1 (GLP-1) and peptide YY (PYY), which are well known to increase satiety [177]. In addition, the formation of ionic and acidic gels may also be responsible for the reduction in hunger and the digestibility of macronutrients [178]. The control of lipid digestion requires protective interfaces surrounding the lipid droplets. A nondigestible gel matrix such as alginate aids in controlling lipase diffusion. Under gastric conditions, the beads contract, swell in the intestine and then disintegrate at the end of the intestinal phase [179]. Thus, alginate is an attractive option for controlling the release of dietary lipids, due to the adaptability of beads to respond to changes in environmental conditions encountered in the gastrointestinal tract [180]. Although the mechanisms by which alginate acts on the pathways that control food intake and the regulation of body weight are not completely elucidated, the evidence thus far suggests that alginate is a promising candidate for the treatment of obesity, with multiple health benefits. Figure 3 depicts the overall effects and the mechanisms by which alginates participate in the management of diabetes and obesity.

## 5. Conclusions and Perspectives

Alginates are natural compounds derived from marine brown algae and have traditionally been extensively used as additives and ingredients in the food industry. However, due to their properties, such as biocompatibility, nontoxicity, biodegradability and functional versatility with various matrices and substrates, alginates have been evaluated for the development of wound dressings. Both in vitro and in vivo results highlight the advantages of using alginate in the treatment of wounds due to its hemostatic and antibacterial properties. Depending on the applicability, these dressings can be presented in various forms, such as wounds, hydrogels, foams or aerogels. Wound dressings are biocompatible, biodegradable and nontoxic, with healing properties, and are used for the management of highly exuding wounds such as surgical wounds, leg ulcers and pressure sores. Foam dressings provide a protective cushioning effect over the wound and maintain hydration so that the appropriate moisture environment promotes epithelialization and healing. Hydrogel dressings are applied as controlled drug delivery or wound healing systems, and their properties can be improved by incorporation with natural substances. Alginate bioaerogels possess certain characteristics that promote their use in thermal insulation, energy storage devices, environmental clean-up, and the medical, pharmaceutical and food industries. Being completely natural, they are much more easily tolerated by the human body, thus avoiding the risk of further complications. Moreover, the positive impact on the environment must be taken into account, since these types of dressings replace the conventional, synthetic ones that are difficult to sort and recycle, with harmful effects on the environment. Given that the medical, pharmaceutical and food industries are large pollutants, the use of these free waste biomaterials is a critical step in mitigating the negative impact on the ecosystem. Finally, alginate gels have been used in tissue engineering applications, as well as in bone and cartilage regeneration.

The use of alginates has been greatly expanded to the development of products used in the management of metabolic disorders such as obesity and diabetes. This cluster of diseases represents an enormous risk to public health, leading in terms of the number of global deaths and disabilities, and is responsible for a host of comorbidities and unprecedented economic burden. Furthermore, what is most alarming is the continuing upward trend in the prevalence of obesity and diabetes in children and adolescents, increasing the risk of morbidity and mortality in adulthood. Therefore, for the past several decades, significant efforts have been devoted to developing strategies and treatments to curb obesity and diabetes. It is well established that diet is the most modifiable risk factor in the prevention of obesity and diabetes. Hence, improvements in diet and the discovery of new food derivatives constitute a key element in dietary formulations in order to maintain a healthy metabolic weight. Advances in biotechnology led to a better understanding of the physiological and metabolic interactions between the host and food ingredients. One of the mechanisms through which diet modulates food intake and controls weight gain is the increase of postprandial satiety, which reduces food consumption. Alginates, like other types of dietary fiber, have been shown to regulate appetite by reducing gastric emptying, nutrient absorption and overall feelings of fullness. These effects have been attributed to alginate’s ability to form gel-like viscous structures with gastric contents and delay intestinal transit. Therefore, alginate-based nano- and microformulations have been used to modify eating behavior and impinge on the mechanisms responsible for obesity and diabetes management. One such mechanism acts through the gut microbiota. Emerging evidence implicates the trillions of microorganisms housed mainly by the large intestine in various chronic diseases ranging from gastrointestinal inflammatory and metabolic conditions to neurological, cardiovascular and respiratory illnesses. This has led to a massive research endeavor in studying how changes in the gut microbiota affect disease outcomes. Thus, the demand for functional foods supplemented with probiotics has dramatically increased, as has the demand for technology to manufacture and deliver food products that retain their sensory characteristics, are stable and provide long-term health benefits. Probiotic viability is affected by several factors that include pH, temperature, presence of oxygen, hydrogen peroxide, and other adverse environmental conditions. Encapsulation using various materials, such as sodium alginate, has a significant protective effect on the stability and viability of gut bacteria, and microencapsulation has been successfully adopted by the food industry. Several methods of encapsulation have been used, employing various vehicles such as beads or capsules with simple or multilayer core-shell structures. However, the survivability and sensitivity of a bacterial strain to various conditions can be different, and therefore other types of encapsulation vehicles and coating layers should be tested under multiple storage and gastrointestinal conditions to accommodate the variability in sensitivity and tolerance of different bacteria. Furthermore, future research should also consider alginate encapsulation with food products that confer increased satiation and decrease caloric intake. Long-term randomized clinical studies should evaluate the effects of alginate on food sensory properties, meal sequencing, gastric emptying, nutrient absorption, secretion of gastrointestinal satiation peptides, as well as on the hedonic and reward systems that control nonhomeostatic eating.

In addition to its role in food intake and obesity management, alginate has been widely tested as a vehicle for drug delivery as well as an antidiabetic agent. Recent advances in the development of nanoparticles encapsulated with various polymeric matrices have allowed for the incorporation of drugs such as GLP-1 and DPP-IV inhibitor in order to permeate the intestinal barrier and control glycemic responses. For example, alginate- and chitosan-based formulations have paved the way for the development of nanoparticles that enable drug delivery, a task that has been a challenge in the management of diabetes. Future work should also focus on developing alginate products with high ionic cross-linking, which results in increased viscosity, prevents glycemic excursions, maintains glucose hemostasis and improves drug delivery. Challenges still remain in the development of these biomaterials, including the transfer of technology, expanding of the manufacturing process, regulatory safety requirements, environmental concerns and consumer acceptance. While the biocompatibility of alginate is well established, its specific effects on human physiology have yet to be thoroughly characterized. Its use as a metabolic disorder complementary treatment requires further data that examines the interactions with specific molecular markers at the proteomic and genomic levels. This, combined with the results from clinical trials, should pave the way for the common use of alginate in medical or pharmaceutical practice beyond metabolic disorders such as obesity and diabetes.

## Figures and Tables

**Figure 1 polymers-12-02417-f001:**
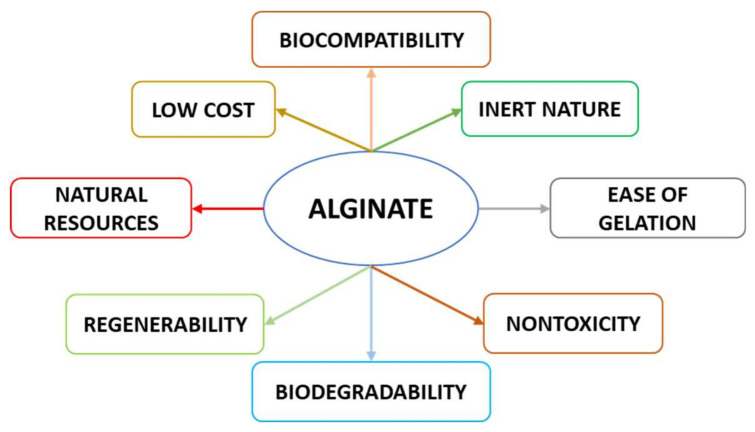
Advantages of alginate.

**Figure 2 polymers-12-02417-f002:**
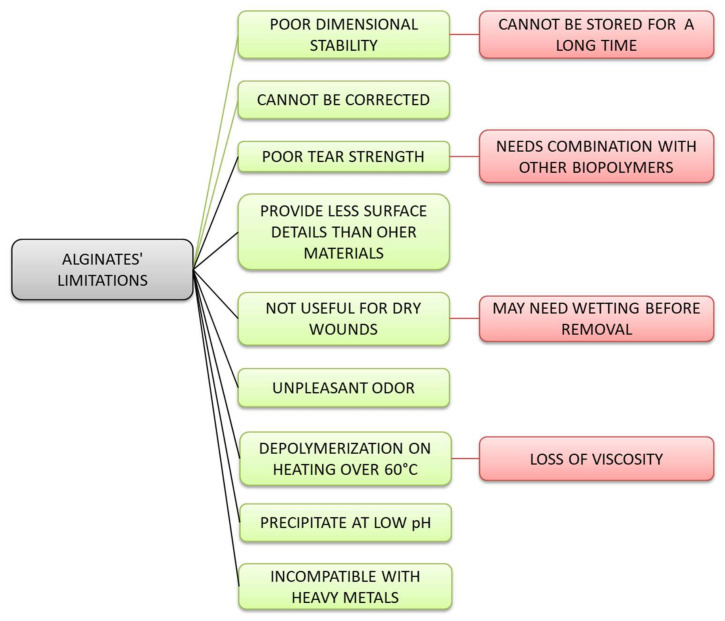
Disadvantages of using alginate in medical applications.

**Figure 3 polymers-12-02417-f003:**
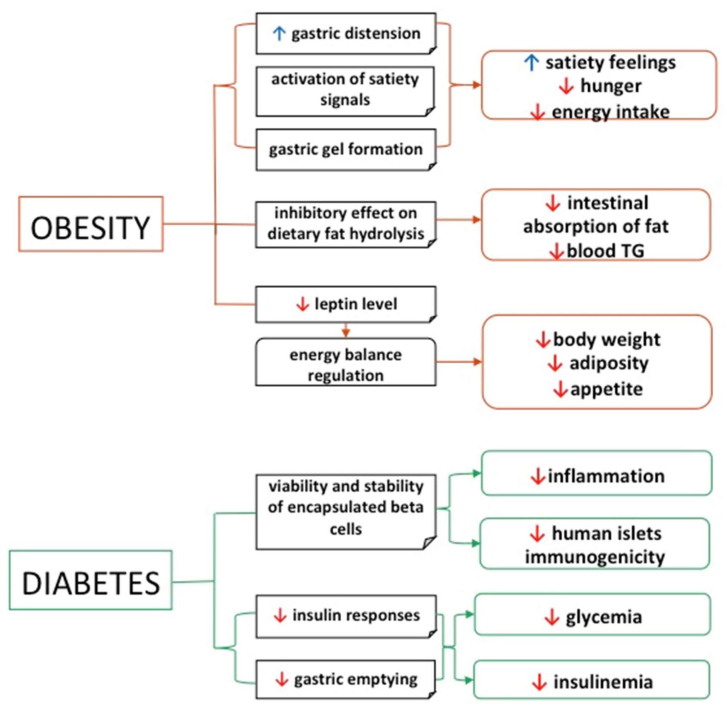
Alginate health benefits, with applications in diabetes and obesity.

**Table 1 polymers-12-02417-t001:** Substances incorporated with alginate.

Product	Substances Incorporated	Effects	References
Chitosan-alginate nanocapsules	turmeric EO and lemongrass EO	Nanocapsules were hemocompatible and used in biomedical and pharmaceutical applications; low and sustained release at neutral pH over 48 h.	[34]
SA and pectin	a-tocopherol	Antioxidant in bakery products. Prevents auto-oxidation and increases the shelf life. Encapsulation facilitates handling, enhances stability and maintains prolonged release.	[35]
SA and guar gum	nisin	Used as material for nisin encapsulation. Bactericidal effect and possibility to be introduced into the food system.	[36]
Alginate films	natamycin	May be used as antimicrobial packaging; are homogeneous, visually attractive, translucent and can be easily processed by different incorporation methods.	[37]
SA	*Eugenia supra-auxillaris* microencapsulation	The encapsulation efficiency was 82%; the microcapsules can be used as food preservatives; maintains the antimicrobial activity against B. subtilis, *B. cereus, P. aeruginosa, S aureus* and *A. niger*.	[38]
Chitosan and SA matrix	clove oil	The emulsion system was stable; separation of the phase occurred after 28 days of storage.	[39]
Calcium alginate-clay beads used for the saccharification of cassava slurry into glucose	multienzymes (alpha-amylase, glucoamylase and cellulase	Under optimal conditions, the immobilization yields and the loading efficiency of enzymes were 97.07%. The beads maintained 51.77% of the residual enzyme’s activity after seven hydrolysis cycles.	[40]
Alginate microcapsules	seasoning EO	Due to bioactive and flavoring properties, the EO microcapsules can be incorporated into functional foods. The safety and the sensorial properties of foods through the addition of natural flavorings and preservatives can be improved.	[41]
Edible alginate film	lemongrass oil	Films with lemongrass oil concentrations of 1250, 2500 and 5000 ppm inhibit growth of *L. monocytogenes* and *E. coli.* Practical application of these films for shelf life extension of fish, meat or cheese.	[42]
Alginate/PVOH capsules	limenone	Stability of encapsulated d-limonene in comparison with free aroma; the mixture alginate-polyvinyl alcohol represents an efficient aroma encapsulation matrix.	[43]
Chitosan and alginate microcapsules	cinnamon EO	Microcapsules were of uniform size, with a sustained release of EO exceeding 168 h.	[44]
SA-Based Green Packaging Films	guava leaf extracts	Enhanced antioxidant and antibacterial abilities of packaging material; the results encourage the use of agricultural byproducts that provide functional ingredients.	[45]
Alginate coating	acerola puree	The alginate-acerola puree coating extended fruit stability by decreasing ascorbic acid and weight loss, decay incidence and by delaying the ripening process.	[46]
Chitosan and SA capsule	linseed oil	Quality of oil increased after encapsulation; chitosan and sodium alginate hydrogel can be used to protect food ingredients stored in aquatic environments such as linseed oil.	[47]
Alginate microspheres	oral DNA vaccine against IHNV	The vaccine reduced the virus incidence in the tissues of vaccinated fish. After the oral administration of increasing concentrations of a DNA vaccine against IHNV, there was a significant increase in fish immune responses and resistance to an IHNV infection.	[48]
Alginate	vitamin D_3_	Liposoluble nutraceuticals are incorporated in alginate nanocapsules, with sustained release in gastrointestinal fluid.	[49]
Zein/caseinate/ alginate nanocapsules	propolis	The bioaccessibility of propolis encapsulated in nanocapsules was improved by 80% compared to free propolis (aprox. 30%).	[50]

EO—essential oil, SA—sodium alginate, PVOH—polyvinyl alcohol, INHV—infectious haematopoietic necrosis virus.

**Table 2 polymers-12-02417-t002:** Effects of alginate in vivo.

Biopolymer	In Vivo Applications	Results	References
Caseinate nanoparticles loaded with DOX coated with alginate	tumor-bearing mice	Nanoparticles facilitated controlled and sustained drug releasing and enhanced DOX effectiveness against Ehrlich carcinoma.	[53]
Alginate hydrogel with cianoside	mouse skin with inflammation or atopic dermatitis	Alginate reduced the number of T cells, mast cells and histiocytes, paw skin, ear tissue inflammation, and inflammatory infiltrates.	[54]
Chitosan/alginate/lovastatin nanoparticles	adult healthy Swiss mice	Formulated as a new drug carrier, the nanoparticles were safe, nontoxic and could be applied to lower serum cholesterol.	[55]
SA-based hydrogel beads with diclofenac sodium	Wistar rats	Good delivery system for drugs that could irritate the stomach, such as diclofenac sodium.	[56]
Buprenorphine-loaded rifampin/polyethylene glycol/alginate nanoparticles	Wistar rats	Decreased drug dose consumption and liver tissue damage.	[57]
Folic acid-grafted solid lipid nanoparticles incorporated in alginate microbeads	Balb/c mice	Coated microbeads released IHT in the colon region next to tumors, with efficiency in treatment of colorectal cancer. Showed antitumor effects against HT-29 cells.	[58]
Alginate microcapsules with Beta-TC-6 cells	diabetic mice	Although the microcapsules restored normoglycemia in diabetic mice, the effects were lost after 35 days.	[59]
Pregabalin alginate-taro corms mucilage microspheres	male albino rabbits	Blended microspheres increased bioavailability and half-life, being an emerging potential pharmaceutical excipient for sustained drug release.	[60]
*Bletilla striata*—SA microspheres	male Sprague-Dawley rats	Good gastroretentive drug delivery system due to strong adhesion to gastric mucosa and long resistance time in the stomach.	[61]
5-HMF and silver nanoparticles incorporated in PVOH/SA hydrogels	male Sprague-Dawley rats	Hydrogel accelerated wound healing, neovascularization, wound closure, promoting re-epithelization and collagen deposition.	[62]
Silk fibroin/SA composite porous materials	male Sprague-Dawley rats	Subcutaneous implantation materials were infiltrated, and, although well tolerated, they largely lost their structural integrity after 21 days.	[63]
Alginate hydrogel with H_2_S as wound dressing material	Wistar rats	Treatment facilitated formation of sebaceous glands, hair follicles and complete epithelialization, without fibroplasia or inflammation.	[64]
β-estradiol and BMP-2 alginate scaffolds	osteoporotic and nonosteoportic rats	Without effect in bone mineralization and bone regeneration process.	[65]
3D bioprinted gelatin SA scaffold	Rat Schwann cells	The construct maintained viability and promoted adhesion of Schwann cells, with good biocompatibility and improved cell adhesion.	[66]
Freeze-gelled alginate/gelatin scaffolds	Wistar rats	Scaffolds contributed to the wound healing process, by collagen synthesis and remodeling, with rejuvenation of hair follicles and skin appendages.	[67]
Exosome—alginate based hydrogel	Wistar rats	The composite enhanced wound closure, re-epithelization, collagen deposition, and angiogenesis at the wound beds.	[68]
Naringenin - alginate hydrogel	Wistar rats	The wounds were almost healed after two weeks.	[69]
PVOH/ SA hydrogel-based scaffold with bFGF-encapsulated microspheres incorporated	Wistar rats	Healing process was accelerated due to epithelialization, collagen deposition and antimicrobial effect, by inhibiting *S. aureus* and *E. coli* growth.	[70]
PLA/PVOH/SA	Sprague-Dawley male rats	Positive effects on collagen deposition, angiogenesis and inflammation, and reduced the inflammatory responses during early wound healing.	[71]
Alginate and growth factors	C57/BL6 mice	The composite promoted the healing process, formation of granulation tissue, new collagen deposition and rapid skin regeneration.	[72]
Alginate/gelatine/silver nanoparticles	adult females Wistar rats	Nanoparticles accelerated tissue formation and promoted earlier development of primary collagen scars.	[73]
Norbornene-modified alginate	female C57/Bl6 mice	Due to its good tissue and cell infiltration process, it can be useful in tissue engineering, such as regeneration and drug delivery.	[74]

DOX—doxorubicin, SA—sodium alginate, HMF—hydroximethylfurfural, PVOH—polyvinyl alcohol, bFGF—growth factors, PLA—polylactic acid.

**Table 3 polymers-12-02417-t003:** Probiotic strains encapsulated with alginate.

Product	Microencapsulated Strains	Characteristics	References
Alginate and gelatin microcapsules	*L. rhamnosus*	The concentration of viable cells decreased with an increase in the concentration of the polymers; cell resistance of *L. rhamnosus* (10^5^ CFU g/L) exceeded four months.	[149]
SA microcapsules	*Bifidobacterium* BB-12	After 120 days of cold storage (−18 °C), 7.31 log CFU g^−1^; stability increased with decrease in temperature.	[150]
Chitosan and alginate beads	*L. lactis ssp. lactis*	Subjected to milk fermentation, coating had a significant effect on the rate of cell release within 50 h of continuous fermentation.	[151]
Edible films based on alginate or whey protein	*B. animalis subsp. lactis* BB-12 and prebiotics (inulin and fructooligosaccharides)	Viability was maintained within the minimum threshold (106 CFU/g) necessary to act as a probiotic during 60 days of storage at 23 °C. Incorporation of prebiotic compounds improved *B. animalis subsp. lactis* BB-12 viability, with inulin showing the best performance; viability was maintained at 7.34 log CFU/g.	[152]
Calcium alginate macrocapsules	*L. casei* DSPV318T and *L. plantarum* DSPV354T	Refrigeration maintained concentration above 10^9^ CFU/capsule until day 70, and storage at −20 °C showed counts above 10^9^ CFU/capsule until the end of the study (84 days).	[153]
Symbiotic chewing gum	*L. reuteri*	After 21 days, the number of *L. reuteri* in the encapsulated probiotic chewing gum was higher than in the free probiotic.	[154]
Starch, chitosan and alginate microencapsulation	*L.* *acidophilus*	Lyophilized microparticles showed values above 6 log CFU g^−1^ at cold and frozen temperatures, counts within the range for probiotics for 60 days of storage.	[145]
Alginate-spheres	*S. enteritidis* phage f3αSE	Encapsulation in alginate-Ca^+2^ spheres extended viability. Used as a phage dosification method in water flow systems (phage concentration 102–104 PFU/mL during 250 h).	[155]
SA, pectin, carrageenan and gelatin edible films	*L. rhamnosus*	Storage stability (over 25 days) of *L. rhamnosus* at both tested temperatures (4 and 25 °C), in descending order, was carrageenan > sodium alginate > gelatin > pectin.	[156]
SA, chitosan and HPMC	*L. acidophilus* NCIMB 701748	Inactivation rates of *L. acidophilus* NCIMB 701748 in powders stored at 25 °C were in the following order: HPMC > control > alginate >>> chitosan.	[143]
Chitosan-coated alginate microcapsules	*B. longum*	Chitosan-coated alginate microcapsules protected *B. longum* from gastrointestinal fluid and high-temperature conditions.	[157]
Probiotic baked cereal (with SA)	*L. rhamnosus* GG	Use of air-dried probiotic sodium alginate film improved viability of *L. rhamnosus* GG under simulated gastrointestinal conditions. A bread slice delivered ~7.57–8.98 and 6.55–6.91 log CFU/portion before and after in-vitro digestion.	[158]
SA—sodium CMC films	*L. lactis*	Films showed significant bacteriostatic activity against *S. aureus* at refrigeration conditions for up to one week.	[159]

SA—sodium alginate, HPMC—hydroxypropyl methylcellulose.

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
