# Peer review of "Alginate: From Food Industry to Biomedical Applications and Management of Metabolic Disorders"

_polymers, 2020, doi:10.3390/polym12102417_

Round 1
Reviewer 1 Report
The manuscript reviews alginate applications, covering their possibilities, and also actual uses, for food, biomedical and pharmaceutical industries. Although alginate applications have been widely reviewed in the past, it is true that this review indicates less commons potential applications, such as obesity and diabetes. Although this manuscript can be published in this journal, mayor and significant revisions have to be done in the manuscript concerning structure, and readability. Moreover, insights and novelties of this review have to be properly highlighted.
Comments:
- Although the structure of a review can differ from the typical structure of a manuscript, all manuscript must include the novelties and the state of the art. Many reviews have been published before, covering alginate applications for tissue engineering, drug delivery, or even more general review, such as Lee and Mooney (Progress Polymer Science 37 (2012) 106-126) or Sun and Tan (Materials 6 (2013) 1285-1309),…. The authors must include an introduction in which the state of the art must be covered, highlighting the novelties of this review with respect to other published reviews.
- Many of the applications that have been explained in the manuscript need a previous processing of the alginate into hydrogels, aerogels, foams, particles, beads, films,…In fact, the alginate is used as a carrier the majority of the times. Therefore, the techniques to process the alginate must be explained (atomization, layer-by-layer, hydrogels production, hydrogels drying,…) and must be discussed by the authors in a different section.
- Although there are some comments for in vivo applications in the manuscript, I would create a new section to discuss more extensively the use of alginate in in vivo That fact can indicate the potential use of alginate in the different applications in future use.
- Since it is a review, the authors must also indicate the cons of alginate uses for the different applications.
- The review includes two tables and one figure. In order to improve the readability of the manuscript, more figures must be included to visually summarize the review (alginate applications, alginate advantages, disadvantages,….)
Author Response
Reviewer #1
The manuscript reviews alginate applications, covering their possibilities, and also actual uses, for food, biomedical and pharmaceutical industries. Although alginate applications have been widely reviewed in the past, it is true that this review indicates less commons potential applications, such as obesity and diabetes. Although this manuscript can be published in this journal, mayor and significant revisions have to be done in the manuscript concerning structure, and readability. Moreover, insights and novelties of this review have to be properly highlighted.
Response: We appreciate the constructive comments that helped improve the manuscript. The paper was extensively revised as suggested. See specific comments below.
Comments:
- Although the structure of a review can differ from the typical structure of a manuscript, all manuscript must include the novelties and the state of the art. Many reviews have been published before, covering alginate applications for tissue engineering, drug delivery, or even more general review, such as Lee and Mooney (Progress Polymer Science 37 (2012) 106-126) or Sun and Tan (Materials 6 (2013) 1285-1309),…. The authors must include an introduction in which the state of the art must be covered, highlighting the novelties of this review with respect to other published reviews.
Response. We have added an Introduction section, where we briefly discussed the current findings related to the topics of the review, highlighting the novelty and the importance of the paper in relation to other published reviews (page 2-3). Several additional references (38) have been added including the ones suggested by the reviewer.
- Many of the applications that have been explained in the manuscript need a previous processing of the alginate into hydrogels, aerogels, foams, particles, beads, films,…In fact, the alginate is used as a carrier the majority of the times. Therefore, the techniques to process the alginate must be explained (atomization, layer-by-layer, hydrogels production, hydrogels drying,…) and must be discussed by the authors in a different section.
Response: As suggested, we have included a section describing the main processing techniques (page 13-15) as well as the methodology for each respective biomedical application (page 15, 16-17, 18)
- Although there are some comments for in vivo applications in the manuscript, I would create a new section to discuss more extensively the use of alginate in in vivo That fact can indicate the potential use of alginate in the different applications in future use.
Response: As requested, we have added a new section titled “in vivo applications” that includes a Table (Table 2) summarizing the main effects of alginate preparations in several in vivo models (page 11-13).
- Since it is a review, the authors must also indicate the cons of alginate uses for the different applications.
Response: We have added two new figures that depict the main advantages and disadvantages of alginate use (Fig 1 and Fig2).
- The review includes two tables and one figure. In order to improve the readability of the manuscript, more figures must be included to visually summarize the review (alginate applications, alginate advantages, disadvantages,….)
Response: We have added one Table (Table 2, in vivo applications) and three new figures (Fig 1, 2 and 3), summarizing advantages, limitations and the effects of alginate and alginate-based products in the management of obesity and diabetes.
Reviewer 2 Report
The manuscript polymers-934701 presented by Puscaselu and coworkers reviews alginate properties and its use in biomedical applications and treatment of metabolic disorders.
Overall, the manuscript suffers from some weaknesses and raises concerns regarding the novelty, significance and impact in the field of Polymers.
A Pubmed search about alginate retrieved 168 review papers alone. So, in my opinion, this manuscript should bring some kind of novelty in terms of the subjects addressed or some new insights in terms of discussion in order to bring significant originality to the field. But, after reading the manuscript, one realizes that it is a compilation of several other reviews and original research on this subject, being too wide in terms of addressed areas (chemistry, food industry, biomedicine, pharmaceutical industry, metabolic disorders) and lacking focus and depth in terms of discussion, thus limiting its originality.
Specific Comments
- Figure 1. If it’s not an original figure it should have a reference.
- The sentence “Alginates have been shown to act on human macrophages by activating the proinflammatory cascade leading to resolving inflammation characteristic of a healing process“ (lines 84-86) is not clear. The activation of the proinflammatory cascade doesn’t resolve inflammation, rather develops it. The sentence should be written more clearly.
- line 320 - is the particle size right? it seems a very high particle size for a nanoparticle… reference 85 indicates a particle size below 250 nm and not 250 µm.
- line 408 - the percentage values should be presented with its respective standard deviation to understand if it has a statistical significance.
- Some references (for instance reference 35) should be replaced by more relevant ones in the respective field, since they are available online.
- The abstract refers to alginate as a potential treatment for diabetes. This is misleading, as this substance is mainly used as a drug delivery system and it can’t be considered as a treatment by itself. Also, the authors refer that a special emphasis in the role of alginate in the treatment of diabetes and obesity, but they only dedicate about four pages of the document to this, in a total of 26 pages (excluding the references). So, it appears that the authors fail to objectively address the questions they are proposed in the abstract.
- The conclusions also don’t accurately reflect the themes addressed in the document. A lot of attention is paid the metabolic disorders in the conclusion, but they are only lightly addressed in the document (as referred previously, only 4 pages were dedicated to this subject). Also, the concept of microbiota is more detailed in the conclusions than in the document itself. I think it should be the other way around.
In conclusion, in the present form, the manuscript does not meet strictly the requirements to be published in this journal and needs a conceptual revision before being considered for publication.
Author Response
Reviewer #2
The manuscript polymers-934701 presented by Puscaselu and coworkers reviews alginate properties and its use in biomedical applications and treatment of metabolic disorders.
Overall, the manuscript suffers from some weaknesses and raises concerns regarding the novelty, significance and impact in the field of Polymers.
A Pubmed search about alginate retrieved 168 review papers alone. So, in my opinion, this manuscript should bring some kind of novelty in terms of the subjects addressed or some new insights in terms of discussion in order to bring significant originality to the field. But, after reading the manuscript, one realizes that it is a compilation of several other reviews and original research on this subject, being too wide in terms of addressed areas (chemistry, food industry, biomedicine, pharmaceutical industry, metabolic disorders) and lacking focus and depth in terms of discussion, thus limiting its originality.
Response. We appreciate Reviewers comments and have extensively revised the manuscript to incorporate the recommendations which improved the manuscript substantially.
Specific Comments
- Figure 1. If it’s not an original figure it should have a reference.
Response: The figure has been removed since it did not add to the relevance of the paper.
- The sentence “Alginates have been shown to act on human macrophages by activating the proinflammatory cascade leading to resolving inflammation characteristic of a healing process“ (lines 84-86) is not clear. The activation of the proinflammatory cascade doesn’t resolve inflammation, rather develops it. The sentence should be written more clearly.
Response: Our apologies for the error, which has now been corrected (page 6)
- line 320 - is the particle size right? it seems a very high particle size for a nanoparticle… reference 85 indicates a particle size below 250 nm and not 250 µm.
Response: Thank you for your vigilance. This was an error which has now been corrected (250 nm) (page 21).
- line 408 - the percentage values should be presented with its respective standard deviation to understand if it has a statistical significance.
Response: The results in the original paper do not include standard deviation or standard error, although it states that the effect was statistically significant. We changed the sentence to read as follows: “The mixture of alginate with chitosan improved blood glucose levels and insulin bioavailability by approximately 8.11%”. (page 25)
- Some references (for instance reference 35) should be replaced by more relevant ones in the respective field, since they are available online.
Response: We have removed reference# 35 and replaced with more relevant references (i.e., #71 for this particular finding). Please note that an additional 38 references have been added in the revised manuscript.
- The abstract refers to alginate as a potential treatment for diabetes. This is misleading, as this substance is mainly used as a drug delivery system and it can’t be considered as a treatment by itself. Also, the authors refer that a special emphasis in the role of alginate in the treatment of diabetes and obesity, but they only dedicate about four pages of the document to this, in a total of 26 pages (excluding the references). So, it appears that the authors fail to objectively address the questions they are proposed in the abstract.
Response: The reviewer is correct, and we would like to thank him/her for the observation. Consequently, we have changed “treatment” with “management”, including in the title, that better reflects the effects of alginate products in control of food intake, energy balance and glucose homeostasis. As for the abstract, considering the suggestions from Reviewer#1 (i.e. expanding the applications of alginate as well as processing techniques) we have tailored the abstract accordingly, to reflect the overall content of the paper.
- The conclusions also don’t accurately reflect the themes addressed in the document. A lot of attention is paid the metabolic disorders in the conclusion, but they are only lightly addressed in the document (as referred previously, only 4 pages were dedicated to this subject). Also, the concept of microbiota is more detailed in the conclusions than in the document itself. I think it should be the other way around.
Response: In the revised manuscript, we have added an Introduction section summarizing current findings pertinent to the topics presented that extend beyond the role of alginate in the management of obesity and diabetes. Similarly, the conclusion/perspective section has been modified to better line up with the overall subject matter of the manuscript.
In conclusion, in the present form, the manuscript does not meet strictly the requirements to be published in this journal and needs a conceptual revision before being considered for publication.
Response: I hope you will find our revision satisfactory for publication in Polymers.
Reviewer 3 Report
Please find attached

Author Response
Reviewer #3
The review article “Alginate in biomedical applications and treatment of metabolic disorders” of Puscaselu et al. covers a broad range of alginate’s applications and discusses recent literature on very important fields of applications such as food industry, wound/foam/hydrogel dressings, tissue engineering and treatment of diabetes/obesity. It is well-written and will offer a very useful reference for the research related to alginate. Therefore, I suggest its publication in Polymers after minor suggestions regarding some additional recent literature:
Response. Thank you for the positive comments on our paper.
- https://doi.org/10.1016/j.foodchem.2017.02.071
Which demonstrates emulsification/internal gelation methods for preparation of particles for the delivery of peppermint phenolic extract.
- https://doi.org/10.1021/jf1020347
For the use of hydrophobic alginates for sustained vitamin D3 release.
- Food Funct., 2019,10, 635-645
Zein/caseinate/alginate nanoparticles for the encapsulation of propolis.
Response: All suggested references have been included and the main findings have been cited accordingly in the paper. The new references are #42, 43 and 70.
Round 2
Reviewer 1 Report
Authors´ responses are satisfactory and the quality of the manuscript is improved. Some of the alginate processing techniques and some in vivo applications are now included. Also an introduction is written with some figures to improve the readability. The article can be published in the journal after the following correction.
- Include references in the second paragraph of the introduction.
Author Response
Thank you for the suggestion. We have now inserted references to the second paragraph of the Introduction section (Ref #10-17).
Reviewer 2 Report
The revised manuscript adresses most of the questions I raised in the previous review.
The only aspect I still have some doubts is regarding novelty, but this is a good effort in terms of comprehensive analysis and summary of currently available data in this field.
Likewise, I consider that this revised edition is ready for publication.
Author Response
We appreciate your constructive comments which have helped improved the manuscript substantially.